# Effect of Ultrasound Combined with Glycerol-Mediated Low-Sodium Curing on the Quality and Protein Structure of Pork Tenderloin

**DOI:** 10.3390/foods11233798

**Published:** 2022-11-25

**Authors:** Sha Gu, Qiujin Zhu, Ying Zhou, Jing Wan, Linggao Liu, Yeling Zhou, Dan Chen, Yanpei Huang, Li Chen, Xiaolin Zhong

**Affiliations:** 1School of Liquor and Food Engineering, Guizhou University, Guiyang 550025, China; 2Key Laboratory of Agricultural and Animal Products Store and Processing of Guizhou Province, Guiyang 550025, China; 3Key Laboratory of Plant Resource Conservation and Germplasm Innovation in Mountainous Region, Ministry of Education, College of Life Sciences, Guizhou University, Guiyang 550025, China

**Keywords:** ultrasound, glycerol, low-sodium, water activity, mediated curing

## Abstract

Considering the hazards of high salt intake and the current status of research on low-sodium meat products, this study was to analyze the effect of ultrasound combined with glycerol-mediated low-sodium salt curing on the quality of pork tenderloin by analyzing the salt content, water activity (aw), cooking loss, and texture. The results of scanning electron microscope (SEM) analysis, Raman spectroscopy, ultraviolet fluorescence, and surface hydrophobicity were proposed to reveal the mechanism of the effect of combined ultrasound and glycerol-mediated low sodium salt curing on the quality characteristics of pork tenderloin. The results showed that the co-mediated curing could reduce salt content, aw, and cooking loss (*p* < 0.05), improve texture and enhance product quality. Compared with the control group, the co-mediated curing increased the solubility of the myofibrillar protein, improved the surface hydrophobicity of the protein, increased the content of reactive sulfhydryl groups (*p* < 0.05), and changed the protein structure. The SEM results showed that the products treated using a co-mediated curing process had a more detailed and uniform pore distribution. These findings provide new insights into the quality of ultrasonic-treated and glycerol-mediated low-salt cured meat products.

## 1. Introduction

Salt is an indispensable seasoning and food preservative in the curing process of meat products, which can improve their texture and sensory properties and give them characteristic flavor [1]. Additionally, it reduces the moisture content of products, lowers aw [2], inhibits microbial growth [3], and plays an important role in food storage [4]. Currently, the salt content of salt-cured products is generally high, and millions worldwide die each year from diseases such as hypertension, coronary heart disease, and stroke caused by excessive sodium intake [5]. Numerous studies have shown that reducing sodium intake can reduce the risk of related cardiovascular diseases [6,7]. The World Health Organization has recommended a reduction in sodium intake to less than 2 g/day (5 g/day of salt) for adults by 2025. Hence, a frenzy of studies on low-salt meat products has ensued in recent years. However, low-salt curing reduces the overall quality of meat products, increases their water activity (aw), and enhances microbial and enzymatic activities [8], and the decrease in salt has a substantial effect on protein solubility in meat products [9].

Mediated curing (MC) refers to the systematic construction of low-sodium curing of meat products using exogenous food additives as a medium or using physical methods by affecting the behavior of salt permeation diffusion pathways and water migration in the matrix [10]. Food-grade glycerin is a colorless, sweet-tasting, and viscous additive that reduces aw and extends the shelf life of food when used with other food additives, such as sorbitol and phosphate complexes [11]. The addition of glycerol significantly increases the antibacterial effect caused by the reduction in aw and has great value in food preservation [12]. The proper amount of glycerol addition not only improves the lipid structure of meat but also enhances its quality characteristics [13]. Liu et al. [14] found that glycerol has a bidirectional regulatory effect on the ability to remove water from the meat and inhibits the diffusion of sodium chloride (NaCl) into the meat during the dry curing process. 

Ultrasound is an innovative green processing technology widely used in the food industry processes, including emulsification, extraction, and physicochemical modification. The principle of action of ultrasound is mainly attributed to the cavitation effect [15]. Ultrasound propagates through the liquid by a series of compressions and expansions generated by the probe, leading to the formation of cavitation bubbles. These bubbles grow to a critical size and then collapse violently. These effects further generate strong shear, turbulence, and cavitation forces, which change the physicochemical and functional properties of the protein dispersion [16]. Compared with traditional techniques, it has the advantages of being highly efficient, instantaneous, safe, environment-friendly, and having low economic cost. Studies have shown that ultrasound treatment (UT) accelerates the diffusion process, shortens the curing time, positively affects mass transfer during the curing process [17,18], reduces the salt content, and improves the quality of low-salt cured meat products [19]. Moreover, UT can improve protein properties by reducing particle size and promoting the de-folding of secondary structures, thus improving product characteristics [20]. Furthermore, these changes may alter the exposure of internal groups, such as hydrophobic amino acids and sulfhydryl groups, and further lead to changes in the binding sites on the protein surface, such as hydrophobic binding and hydrogen bonding sites [21].

However, ultrasound and glycerol improve the quality of low-sodium meat products to a limited extent, and there are few studies on UT combined with glycerol-mediated curing. This study aimed to reveal the effect of ultrasound combined with glycerol-mediated low-sodium curing (co-mediated curing) on the quality characteristics of pork tenderloin. Additionally, this work investigated the effect of co-mediated curing on the structural and functional properties of myofibrillar proteins (MPs) to explore the mechanism of action.

## 2. Materials and Methods

### 2.1. Materials

Pork longissimus dorsi muscle (Guizhou local pig breed, China) was provided at 24 h postmortem by Huimin Fresh Supermarket in Huaxi District (Agricultural Investment Huimin Fresh Management Co., Ltd., Guizhou, China). Food-grade NaCl was purchased from the local market (Chongqing Hechuan Salt Chemical Co., Ltd., Chongqing, China). Food-grade glycerol was purchased from distributors (Tyco Palm Chemical Co., Ltd., Zhangjiagang, China). Other chemicals used were of analytical grade, such as 5,5′-Dithiobis (2-nitrobenzoic acid) (DTNB) and 8-anilino-1-naphthalenesul-fonic acid (ANS) were obtained from Sinopharm Chemical Reagent Co., Ltd. (Shanghai, China).

### 2.2. Treatment of Pork Loin and Preparation of MP Extracts

After removing the fascia and connective tissue from the surface of the pork loin, it was cut into 50 × 40 × 10 mm^3^ pieces (50 ± 1 g), followed by the addition of 3% salt and different concentrations of glycerol (0, 1%, and 4%) (denoted as C, G1, and G4, respectively), and then vacuum-packed. These samples were placed in a beaker containing an appropriate amount of ultrapure water and treated using ultrasound (Ultrasonic Homogenizer SCIENTZ-IID, Ningbo Xinzhi Biotechnology Co., Ltd., Zhejiang, China) (denoted as U, UG1, and UG4, respectively). The ultrasound probe (Φ = 6) was immersed in the water, 2 cm from the meat surface. The meat was treated at a fixed ultrasound frequency of 25 kHz and 320 W power for 30 min (10 s of ultrasound with 5 s intervals) at a strictly controlled temperature of less than 10 °C. After sonication, all samples were marinated at 4 °C for 24 h.

Myofibrillar protein extraction was performed by referring to the method reported by Jiang et al. [22]. The fat and connective tissues were removed, and approximately 50 g of meat was homogenized in 4 volumes (*w*/*v*) of protein extraction buffer (0.1 mol/L NaCl, 2 mmol/L MgCl_2_, 10 mmol/L sodium phosphates, and 1 mM EGTA, pH 7.0) for 1 min. The homogenate was centrifuged at 2000× *g* for 15 min. The supernatant was discarded, and the procedure was repeated twice. Next, the precipitated samples were washed thrice with 4 volumes (*w*/*v*) of protein washing buffer (0.1 mol/L NaCl) under the same conditions. Finally, the suspension was filtered through four layers of gauze, adjusted to pH 6 with 0.1 mol/L HCl, and then centrifuged. The extracted MP was dissolved in 15 mmol/L piperazine-N, N′-bis (2-hydroxypropanesulfonic acid) buffer (pH 6.25) for subsequent use, and the MP concentration was determined using the bicinchoninic acid method.

### 2.3. Measurement of Quality Traits

The salt content was measured by weighing 1 g of meat sample diluted 10 times with deionized water, homogenized, and measured using a digital salinity meter (ES-421, ATAGO, Tokyo, Japan) [23].

The aw was measured using a previously described method with some modifications [24]. Five milligrams of meat samples were weighed and measured using an aw measuring instrument (Huake HD-4B, Wuxi, China).

To measure the cooking loss, tenderloin samples (5 g) were immersed in an 80 °C water bath for 20 min and then quickly placed in flowing water, cooled to about 24 °C, surface water drained off, and weighed. The weight of the sample before cooking (M_1_) and after cooking (M_2_) was recorded, and the following formula was used to calculate the percentage of cooking loss value [25]:$$\mathrm{Cooking}\;\mathrm{loss}\;\left(\%\right)=\frac{M_{1}-M_{2}}{M_{1}}\times 100$$

The samples were cut into cubes (10 × 10 × 10 mm^3^) and then subjected to texture profile analysis (TPA) using a TOUCH texture analyzer (Bao Sheng Technology Co., Ltd., Shanghai, China). The probe used was TA/36 (36 mm diameter column probe), and the test conditions used were: test speed: 1 mm/s, interval time: 2 s, target mode, and value: deformation 50%, and contact point type and value: load 5 gf [26].

### 2.4. Low-Field Nuclear Magnetic Resonance (LF-NMR)

To analyze the degree of binding and distribution of water molecules in pork loin under different curing conditions, the transverse relaxation time (T_2_) of the samples was determined using an LF-NMR analyzer (NMI20-040 VI, Niumag Analytical Instrument Corporation, Suzhou, China). The samples (1 g) were placed in a vial and then in an NMR tube for analysis [27]. The Carr-Purcell-Meiboom-Gill pulse sequence parameters for T_2_ measurements were set to SF = 22 MHz, ns = 8, TW = 3000 m, and NECH = 2000.

### 2.5. Solubility

The solubility of the samples was determined using the method described by Zhao et al. [28] with slight modifications. The above protein solution was diluted to 5 mg/mL (pH 6.25), centrifuged at 10,000× *g*, 4 °C for 30 min, and the protein concentration of the supernatant was determined using the method mentioned in Section 2.1. Solubility is defined as the percentage of protein concentration in the supernatant before centrifugation relative to the protein concentration in the protein solution. The assay was repeated five times.
$$\mathrm{Protein}\;\mathrm{solubility}\;\left(\%\right)=\frac{\mathrm{Protein}\;\mathrm{content}\;\mathrm{insupernatant}\;\mathrm{solution}}{\mathrm{Total}\;\mathrm{protein}\;\mathrm{content}\;\mathrm{in}\;\mathrm{protein}\;\mathrm{suspension}}\times 100$$

### 2.6. Reactive Sulfhydryl Groups (R-SH)

The reactive sulfhydryl groups in the samples were determined according to the method described by Kang et al. [29] with modifications. The protein solution was diluted with phosphate buffer (0.6 M NaCl, pH 6.5) to contain 1 mg/mL protein. Next, 4 mL of diluted protein solution was added to 50 μL of DTNB solution, mixed well, and then reacted thoroughly for 20 min under the light. The absorbance (A_412_) was measured at a wavelength of 412 nm using an enzyme marker (SpectraMax 190, Molecular Devices, Sunnyvale, CA, USA). The assay was repeated three times. The sulfhydryl content was calculated using the following equation:$$R–\mathrm{SH}\;\mathrm{content}\;\left(\mu \mathrm{mol}/100\;\mathrm{mg}\right)=\frac{A_{412}\times 10^{5}\times D}{13,600\times C_{0}}$$
where A_412_: absorption; D: dilution multiple; C_0_: protein concentration.

### 2.7. Surface Hydrophobicity (S_0_-ANS)

The surface hydrophobicity of samples was determined according to the method of Zhang et al. [30] with some modifications. The protein solution was diluted with phosphate buffer (0.6 M NaCl, pH 6.5) to contain 1 mg/mL of protein, and 2 mL of protein dilution was added to 10 μL of 15 mM ANS solution, mixed thoroughly for 5 min, and left at room temperature (20 °C) for 20 min. The fluorescence intensity (a.u.) was measured at the excitation and emission wavelength of 380 and 470 nm, respectively, using an ELISA. The assay was repeated three times.

### 2.8. Raman Spectra Measurements

A Raman spectrometer/microscope (Lab RAM HR EVO, Horiba Jobin Yvon S.A., Paris, France) with a 532 nm laser source was used to collect the Raman spectra. The laser was focused on the sample with a 50 mm long focal length lens before acquiring Raman signals in the 400–2000 cm^−1^ range. The measurement parameters were 2 cm^−1^ resolution, 60 s exposure time, and five scans for each sample. To remove the fluorescent background, the spectra were smoothed using Labspec6 analysis software and multi-point baseline correction. The spectra were normalized to the vibrational intensity of the phenylalanine ring at a wave number of 1003 cm^−1^ [31].

### 2.9. UV Second Derivative Absorption Spectra

A UV spectrophotometer was used to measure the UV absorption spectra of MP under various treatments in the range of 200–400 nm [32]. The scan accuracy of the UV absorption spectrometer was set to 10, the sampling interval to 1 nm, and the corresponding buffer was used as a blank. The second derivative absorption spectrum was obtained using the OriginPro 2021 program. The letter ‘r’ represents the peak-to-valley ratio of the two main peaks.

### 2.10. Scanning Electron Microscope (SEM)

The microscopic structures of MP samples were determined according to the method described by Wu et al. [33] with slight modifications. The samples were cut into blocks of 2 × 5 × 5 mm^3^ perpendicular to the direction of the myofibers, fixed with 2.5% glutaraldehyde for 48 h at room temperature, washed with 0.1 M phosphate buffer (pH 7.0), and then eluted twice with gradients of 25%, 50%, 75%, and anhydrous ethanol, respectively, each time for 1 h. Next, they were lyophilized for 48 h, plated with gold, and then observed and photographed using an SEM (COXEM EM-30, Seoul, Korea) at an accelerating voltage of 20 kV and a magnification of 300X.

### 2.11. Statistical Analysis

Each experiment was performed in triplicate. Data analysis and visualization of correlations were performed using Origin and Statistical Products and Services Solutions (SPSS) statistical software (version 20.0), and the results were presented as mean ± standard deviation (SD). One-way analysis of variance (ANOVA) and Duncan’s test (*p* < 0.05) were performed to determine differences between samples.

## 3. Results and Discussion

### 3.1. Quality Indicators

#### 3.1.1. aw

As summarized in Table 1, the differences in aw between the groups G4, U, UG1, and UG4 were significant (*p* < 0.05) compared to group C. As the addition of glycerol increased, a more significant decrease in aw was observed. The lowest aw was observed in the UG4 group. This observation was consistent with the study results of Liu et al. [24], reporting that glycerol-mediated curing decreased the aw of ground pork, probably because glycerol contains three hydroxyl groups that could bind to proteins and fats to increase the polarity of certain groups and convert some free water into bound water, thus decreasing aw. The decrease in aw by UT can be attributed to the mechanical effect of microbubble rupture, which increases NaCl transfer and decreases aw [34]. Thus, co-mediated curing was most effective in reducing aw, showing the synergistic effect of both methods during the curing process.

#### 3.1.2. Salt Content

During the meat curing process, the substances are continuously exchanged, and the salt and water contents constantly change due to osmotic pressure. As summarized in Table 1, the salt content of the U group was significantly higher than those of the other groups. Ultrasound treatment could promote salt diffusion and shorten the curing time, which could be attributed to the cavitation effect-induced ultrasonic waves, causing the microchannels and micro vibrations at the object interface to improve the diffusion mechanism and promote salt transportation [35]. The salt content decreased with the increase in the added glycerol amount and significantly differed (*p* < 0.05) from the control group when the amount of glycerol added was 4%. Notably, the salt content was significantly reduced in the ultrasound combined with different glycerol concentration-mediated curing groups. This could be attributed to the increasing glycerol viscosity, leading to osmotic resistance of salt in the meat and hindering salt diffusion [36]. The hydroxyl group of glycerol combines with the hydroxyl group in water through hydrogen bonds, transforming free water into bound water and thus inhibiting salt diffusion. In addition, the glycerol molecules can cause ultrasound-induced microchannel congestion during the curing process, resulting in slow salt diffusion.

#### 3.1.3. Cooking Loss

Regarding the effect of mediated curing on the cooking loss of pork tenderloin, the G1 group did not show statistically significant differences compared to the C group. However, the cooking loss in the G4 group was significantly reduced than in the control group. Glycerol, as a polyol, poses a good water retention capacity. Its three hydroxyl groups combine with water molecules to form hydrogen bonds, thus absorbing large amounts of water and trapping them in a three-dimensional network, which significantly improves the water-holding capacity of the product [14]. The ultrasound-mediated curing reduced the cooking loss of the product probably because UT disrupted the muscle structure of the meat, promoted the dissolution of salt-soluble proteins on the meat surface, prevented water runoff, and improved water retention, which was consistent with the findings of Kang et al. [37]. In the present study, the samples that underwent co-mediated curing had lower cooking losses than the other groups. This result can be attributed to the presence of NaCl, showing denser swelling spaces between the muscle fibers and the microchannels created by the UT, inducing a uniform distribution of glycerol in the sample. With the penetration of glycerol, the charged particles bound to the myofibrils induce electrostatic shielding and increase electrostatic repulsion between myofibril filaments, leading to swelling of myofibrils and retention of more water, thereby improving the water-holding capacity of the product.

#### 3.1.4. Analysis of Texture Parameters

As summarized in Table 1, among the six treatment groups, there were no significant differences in the hardness values, chewiness, and elasticity between the U and C groups (*p* > 0.05), while the elasticity, chewiness, and hardness of the meat showed a significant reduction in the G1, G4, UG1, and UG4 groups, with the lowest value being found in the UG4 group. Sorapukdee et al. [38] found that glycerol curing led to the disruption of cellular structure and altered the MP structure integrity, thereby improving the tenderness of pork tenderloin. Another study by Yeung et al. [39] showed that the treatment of the meat with appropriate ultrasonic intensity reduced its hardness values. This could be attributed to the formation of local hot spots during bubble collapse, cavitation-induced shear, and shock waves [40]. Furthermore, the hardness values observed in this study were lower in the UG groups than in the other groups, confirming the synergistic effect of ultrasound-assisted penetration of glycerol solution on the meat mass. Overall, the co-mediated curing reduced the hardness value and promoted meat tenderization.

### 3.2. LF-NMR

LF-NMR was used to determine the distribution and binding of water in the samples. As depicted in Figure 1a, three peaks were observed in the T_2_ spectrum. T_2b_ (1–10 ms) indicates that the water is tightly bound to the macromolecules or protons located on macromolecular structures plasticized by water; T_21_ (10–100 ms) is mainly the water located within the dense myofibrillar protein matrix; T_22_ (100–1000 ms) is the free water present outside the myofibrils [41].

The ratios of bound water, immobilized water, and free water are depicted in Figure 1b. The addition of glycerol and UT showed different degrees of reduction in T_21_ and T_22_, indicating that glycerol or ultrasound-mediated curing reduced the free water content and increased the bound water, which could affect the water-binding ability of molecules within the system. Co-mediated curing significantly decreased T_21_ and T_22_, probably due to the structural changes in the myofibrils caused by the action of ultrasound, and the further action of glycerol impeded salt diffusion, resulting in the weakening of electrostatic repulsion within the myofibrils, transformation of immobilized water to bound water, and a decrease in free water content [42]. This conclusion also explains the changes observed in aw, as mentioned in Section 3.1.1 (Table 1).

### 3.3. Solubility

As the most abundant protein in muscle, the solubility of MP can directly reflect the functional properties of the protein. The effect of different treatments on the solubility of MPs in porcine loin is depicted in Figure 2a. The groups containing glycerol and treated with ultrasound exhibited higher solubility than the control group, with the greatest solubility observed in the co-mediated group. The solubility of MPs increased with increasing glycerol concentration, mainly because glycerol could localize the protein molecules to the protein surface, forming an additional hydrophilic layer that acted as an amphiphilic interface between the hydrophobic surface and the polar solvent to improve protein stability and increase protein hydration. Preferential interactions between the molecules on the protein surface, in the solvent, and the solution surrounding the protein can alter the protein’s structural stability [43]. Ultrasound treatment can increase protein solubility by its cavitation effect, exposing the internal hydrophilic groups of proteins and causing conformational changes and the formation of soluble protein aggregates. Previous studies have reported that disrupting hydrophobic interactions due to UT-induced turbulence and shear can promote the intermolecular association between the protein molecules, thereby increasing the solubility [44]. Thus, co-mediated curing enhances protein solubility, probably due to the improved unfolding of protein molecules in the presence of ultrasound, which promotes the exposure of internal groups and facilitates the interaction of polyhydroxy groups in glycerol with proteins, leading to increased solubility.

### 3.4. R-SH Content

The R-SH can form disulfide bonds by oxidizing two cysteine residues on the adjacent protein chains and are often regarded as essential indicators for assessing the degree of protein denaturation and refolding and a key factor in maintaining the tertiary and quaternary structure of proteins [45].

Figure 2b depicts the changes in the R-SH content after different curing methods. Moreover, the R-SH content of the G4, U, UG1, and UG4 groups significantly increased than the control group (*p* < 0.05). The increase in the R-SH content caused by the addition of glycerol could be attributed to the non-uniform distribution of glycerol on the surface of the protein molecules, with several hydroxyl groups preferentially interacting with the protein, leading to the exposure of the sulfhydryl groups. The R-SH content significantly increased after UT (*p* < 0.05), and the results suggested that UT caused MPs to unfold, exposing the bound sulfhydryl groups to the molecular surface. Jiang et al. [22] found that the free sulfhydryl group and the total sulfhydryl group contents in the ultrasound-treated samples were significantly higher than those in the control samples due to subunit dissociation, disulfide bond breakage, and exposure of internal sulfur hydroxyl groups caused by protein unfolding. The significant increase in the R-SH content in the co-mediated curing groups could be attributed to the interaction between the polyhydroxy groups in glycerol and protein molecules facilitated by their conformational change and subsequent disulfide bond breakage during the UT process.

### 3.5. S_0_-ANS

S_0_-ANS is an important structure-related factor affecting protein function, such as the surface properties of proteins associated with the exposure of hydrophobic amino acid residues on protein molecules’ surface [46]. As depicted in Figure 2c, no significant difference was observed between the C and G1 groups (*p* > 0.05), and the S_0_-ANS was significantly higher in the G4 group than in the C group (*p* < 0.05). These results suggested that the addition of 4% glycerol caused the exposure of hydrophobic amino acid residues, and these hydrophobic groups were aggregated together with hydrophobic interaction, leading to an increase in surface hydrophobicity. The S_0_-ANS of the U group also showed significant differences from the C group (*p* < 0.05). The results indicate that UT disrupts the hydrogen bonds between the protein molecules and electrostatic interactions and hydration, rearranges large primary protein aggregates, relocates some bound hydrophobic groups to the surface, and finally forms a new surface, following which the hydrophobic core migrates outward from the protein interior, enhancing the binding of MPs to ANS [47]. The combination of ultrasound and glycerol-mediated curing significantly increased S_0_-ANS, which could be attributed to the exposure of ultrasound-induced hydrophobic regions, which could compensate for the embedded glycerol molecules, increase the reaction ability of protein and glycerol, and lead to more significant changes in surface hydrophobicity. 

These results correspond to the increased sulfhydryl groups, confirming that the physical modification induced by mediated curing using ultrasound and glycerol could lead to conformational changes in MPs.

### 3.6. Secondary Structure

Raman spectroscopy can provide information on the conformation of the protein-peptide backbone, amino acid side chains, and the microenvironment around the peptide chain. It can be used to analyze the protein’s secondary structure changes and explore the relationship between protein conformation and quality changes [48].

Figure 3 depicts the Raman bands (Figure 3a) and the relative content of the secondary structures (Figure 3b) of the groups that have undergone different curing methods after baseline calibration and normalization. The results showed that the addition of glycerol and the ultrasound-mediated curing methods led to the transition of α-helix to β-sheet in different degrees, respectively. This might be due to the unfolding of the protein molecules, where the protein structure changes from an ordered to a disordered state due to the disruption of the protein molecules, partial weakening of hydrogen bonds, and formation of β-turn and random coil [15]. Co-mediated curing further promoted the structural changes in MPs, probably because UT exposed more protein surface area, changing the spatial entanglement of protein molecules, structure, and proteolysis, thus improving the ability of these proteins to bind glycerol, enhancing ionic and hydrogen interactions between the protein molecules, and preventing protein denaturation [49]. Generally, the secondary structure of MPs is closely related to the changes in surface hydrophobicity [50]. The exposure of the hydrophobic groups leads to a decrease in intramolecular hydrogen bonding and an increase in surface hydrophobicity.

### 3.7. UV Second Derivative Absorption Spectra

The second derivative of the UV spectrum between 280 and 300 nm was used to illustrate the subtle differences in the amino acid microenvironment. As depicted in Figure 4, the peak at 296 nm was caused by the tryptophan residues, while a combination of tryptophan and tyrosine resides caused that at 287 nm. The distance between the first peak and the first valley is considered ‘a’ while the distance from the second peak to the second valley is considered ‘b’. The ‘r’-value (r = a/b) indicates the ratio of peak to trough position. The higher the ‘r’-value, the greater the surface exposure of tyrosine residues [51]. The ‘r’-values for all mediated cured samples were higher than the control values, indicating that the tyrosine and tryptophan residues were exposed to a hydrophobic environment, which might be due to proteolytic depolymerization. The increase in ‘r’-values indicated that mediated curing led to a change in protein conformation and the exposure of bound tyrosine residues within the MPs to the surface due to protein unfolding. Additionally, UT could expose more hydrophobic amino acids to the surface of the protein molecule, at which point the increased hydrophobic interactions caused the protein to re-condense and re-embed some hydrophobic amino acids [52]. Under UT conditions, the interaction between glycerol and protein molecules was altered due to the change in protein conformation and subsequent bond formation.

### 3.8. SEM

As depicted in Figure 5a, the MPs in the control group showed a loose and rough structure, probably due to the electrostatic interactions caused by low ionic strength to form myosin filaments [53]. There was no significant change in the microstructure of MPs in the group with 1% glycerol addition (Figure 5b) compared to the control group. The experimental group with 4% glycerol addition (Figure 5c) showed smaller myofibril diameters, indicating an increased myofibril density with increasing glycerol concentration. After UT (Figure 5d), the surface of the myofibrils was flat and uniform, but the structure was looser, which might be related to the structural changes in the protein. Moreover, the changes in the pore ratio after UT could be attributed to the strong cavitation effect of ultrasound and its ability to induce protein de-folding [54]. Notably, the ultrasound-mediated glycerol-cured MPs showed the most delicate and tight distribution, and the UG4 group (Figure 5f) samples were relatively smoother, with smaller particles, and were neatly arranged. Therefore, it can be inferred that the entanglement of proteins with the hydroxyl groups in the surrounding environment after UT provided a stable environment for improving the protein content. This improved microstructure can be attributed to the synergistic effect of glycerol and ultrasound treatment.

## 4. Conclusions

In this study, the addition of glycerol and UT reduced aw, cooking loss, and hardness values in the pork loin samples to different extents The LF-NMR results showed that the free water content of the samples was reduced, the immobilized water was converted to bound water, and the water holding capacity of the samples was improved. UT changed the structure of MPs, while glycerol changed their conformational stability through major molecular forces, such as hydrophobic and electrostatic interactions. Under co-mediated curing conditions, the Raman spectroscopy results showed a shift from α-helix to β-sheet, indicating the unfolding of the protein’s secondary structure. This promoted proteolysis at low ionic strength and allowed more hydrophobic groups to be exposed and the content of reactive sulfhydryl groups to increase, leading to hydrophobic aggregation of MP and the formation of a dense network structure. Notably, co-mediated curing has some potential in developing low-salt meat products due to its ability to improve their quality and protein structure.

## Figures and Tables

**Figure 1 foods-11-03798-f001:**
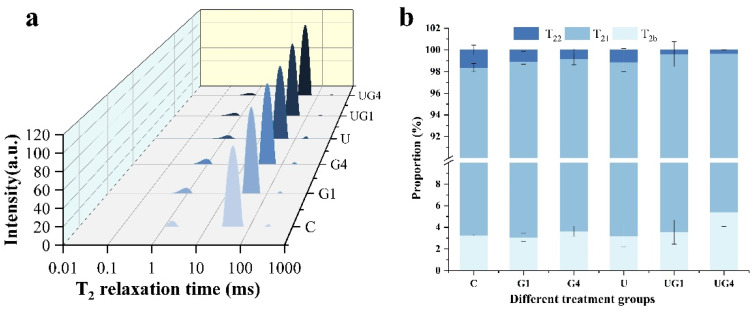
The curve of T_2_ relaxation time (**a**), T_21_, T_2b_, and T_22_ relaxation proportion (**b**) of different treatment conditions. C: 3% NaCl; G1: 3% NaCl and 1% glycerol; G4: 3% NaCl and 4% glycerol; U: 3% NaCl with UT; UG1: 3% NaCl and 1% glycerol with UT. UG4: 3% NaCl and 4% glycerol with UT.

**Figure 2 foods-11-03798-f002:**
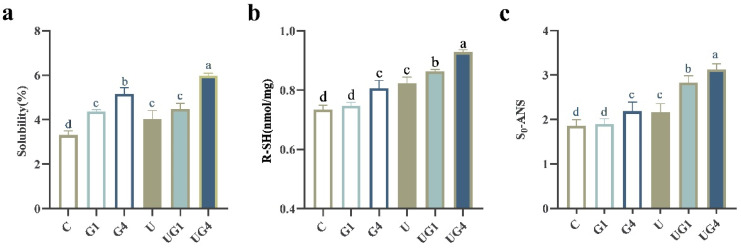
The solubility (**a**), reactive sulfhydryl groups (R-SH) (**b**), surface hydrophobicity (So-ANS) (**c**) under different treatment conditions. ^a–d^ means the significant differences (*p* < 0.05). C: 3% NaCl; G1: 3% NaCl and 1% glycerol; G4: 3% NaCl and 4% glycerol; U: 3% NaCl with UT; UG1: 3% NaCl and 1% glycerol with UT. UG4: 3% NaCl and 4% glycerol with UT.

**Figure 3 foods-11-03798-f003:**
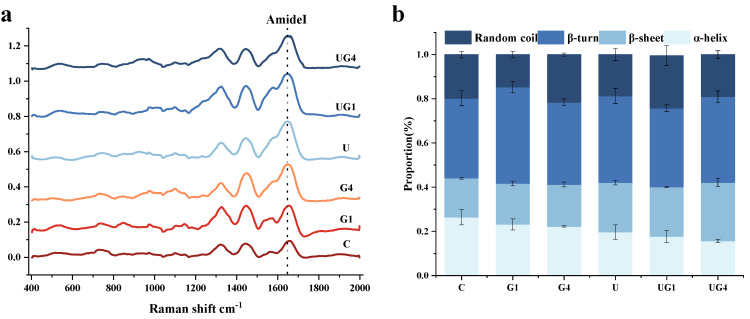
Raman spectra (**a**) and the corresponding secondary structure under different treatment conditions (**b**). C: 3% NaCl; G1: 3% NaCl and 1% glycerol; G4: 3% NaCl and 4% glycerol; U: 3% NaCl with UT; UG1: 3% NaCl and 1% glycerol with UT. UG4: 3% NaCl and 4% glycerol with UT.

**Figure 4 foods-11-03798-f004:**
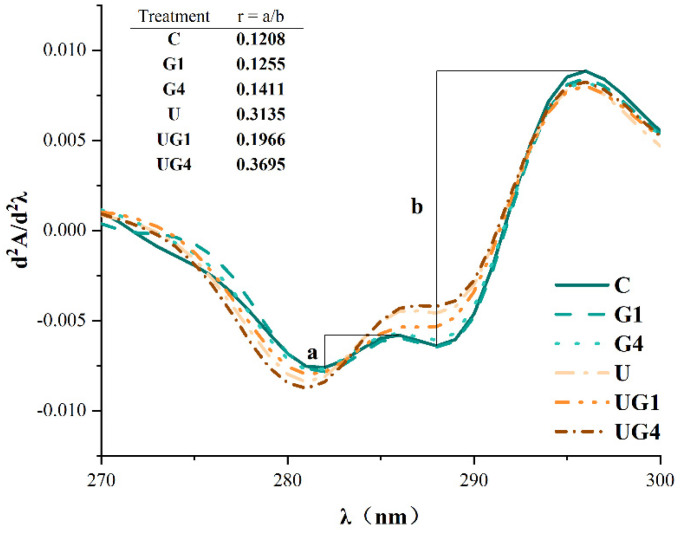
The second derivative of the UV spectra under different treatment conditions. C: 3% NaCl; G1: 3% NaCl and 1% glycerol; G4: 3% NaCl and 4% glycerol; U: 3% NaCl with UT; UG1: 3% NaCl and 1% glycerol with UT. UG4: 3% NaCl and 4% glycerol with UT.

**Figure 5 foods-11-03798-f005:**
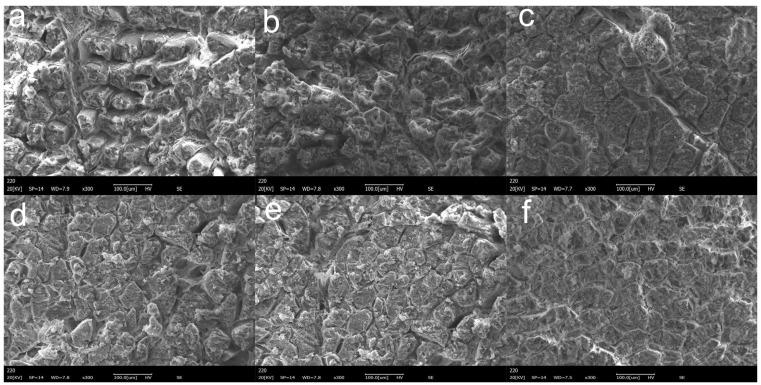
Microstructural changes of different treatment conditions. (magnification 300×). (**a**) 3% NaCl; (**b**) 3% NaCl and 1% glycerol; (**c**) 3% NaCl and 4% glycerol; (**d**) 3% NaCl with UT; (**e**) 3% NaCl and 1% glycerol with UT. (**f**): 3% NaCl and 4% glycerol with UT.

**Table 1 foods-11-03798-t001:** Aw, salt content, cooking loss, and TPA under different treatment conditions.

Samples	aw	Salt Content	Cooking Loss (%)	TPA		
				Hardness (gf)	Chewiness (gf)	Springiness
C	0.967 ± 0.007 ^a^	0.277 ± 0.006 ^b^	19.233 ± 1.150 ^a^	8406.980 ± 397.972 ^a^	3421.360 ± 145.948 ^a^	0.902 ± 0.478 ^a^
G1	0.962 ± 0.002 ^ab^	0.267 ± 0.006 ^bc^	18.410 ± 0.577 ^a^	7129.688 ± 235.097 ^bc^	2784.278 ± 278.093 ^ab^	0.893 ± 0.465 ^a^
G4	0.954 ± 0.003 ^bc^	0.263 ± 0.006 ^c^	16.307 ± 0.670 ^b^	6751.389 ± 125.456 ^cd^	2318.975 ± 309.291 ^bc^	0.584 ± 0.496 ^b^
U	0.952 ± 0.004 ^cd^	0.327 ± 0.006 ^a^	16.973 ± 0.468 ^b^	7555.964 ± 458.378 ^a^	2885.084 ± 321.206 ^ab^	0.682 ± 0.163 ^ab^
UG1	0.946 ± 0.002 ^d^	0.263 ± 0.006 ^c^	16.746 ± 0.783 ^b^	6176.572 ± 370.174 ^de^	2284.247 ± 512.911 ^bc^	0.760 ± 0.196 ^ab^
UG4	0.944 ± 0.002 ^d^	0.243 ± 0.006 ^d^	14.637 ± 0.389 ^c^	5575.223 ± 415.194 ^e^	1736.976 ± 445.675 ^c^	0.547 ± 0.056 ^b^

^a–e^ means the significant differences (*p* < 0.05) in the same column under different treatment conditions. C: 3% NaCl; G1: 3% NaCl and 1% glycerol; G4: 3% NaCl and 4% glycerol; U: 3% NaCl with UT; UG1: 3% NaCl and 1% glycerol with UT. UG4: 3% NaCl and 4% glycerol with UT.

## Data Availability

Data are contained within the article.

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
