# Peer review of "Effect of Ultrasound Combined with Glycerol-Mediated Low-Sodium Curing on the Quality and Protein Structure of Pork Tenderloin"

_foods, 2022, doi:10.3390/foods11233798_

Round 1

Reviewer 1 Report

Hi dear

This article "Effect of ultrasound combined with glycerol-mediated curing on the quality and protein structure of pork tenderloin” was revised and has a novelty and I recommend it after consideration of the following comments.

Title: it is perfect and comprehensive. If it is possible the “low- sodium” add in title.

Abstract:

·       The type of statistical design used in this research should be mentioned.

·       Please include a background of your study in the first of abstract.

·       The results must be expressed as statistical levels i.e., p<0.05 or 0.01 etc.

Introduction:

·       Please pointed the new application ultrasound on food industry as a new paragraph for example you can use this article and so on: (https://doi.org/10.3390/
foods11203263).

·       Please consider the treatments in the final paragraph of introduction.

·       Line 79-80: “their effect” the effect of ultrasound combined with glycerol-mediated low-sodium salt….. Please provide a good abbreviation for “ultrasound combined with glycerol-mediated low-sodium salt” and apply through the manuscript text.

·                 Please write materials as (company, City, Country), especially for chemical analysis assessment which used in the study.

·       Line 114-126: Some methods do not have references, etc.

·       Line 93: Why do you report 3% salt as low sodium? And why high sodium levels were not used in your research? What is the reason for using this amount of salt and glycerol?

·       Table 1: please include the other TPA parameters i.e., gumminess, adhesiveness, adhesive force, and cohesiveness. And please correct the word chewiness.

·       Fig 2C: please self-explanatory for figs trough the tables and Figs for example S0-ANS? Etc.

·       Fig 4: explain in caption a….c.

Discussion text must grammar improve and in some cases it is very weak and maybe there is no discussion at all.

Conclusions:

Conclusion is very general, try to make it more scientific, comprehensive and concise in detail, especially.

Line 407: “the water holding capacity of the samples improved” (Line 407) does it contradict with the addition of glycerol and UT reduced aw (Line 404)?

References: It is OK.

The article has many flaws in express and concept of English, it is suggested to be revised in a scientific and native way.

Reviewer 2 Report

The authors analyzed the effect of ultrasound in combination with low-sodium curing with glycerol on the quality features of pork loin, especially the structural and functional properties of myofibrillar proteins. The undertaken subject is very important nowadays due to the health-promoting aspect of pork and the improvement of technological properties of this raw material. The experiment was planned and carried out correctly. However, the methodology requires minor additions.

Detailed comments and suggestions of the Reviewer can be found below:

Lines 92-94 say that 3% salt and different concentrations of glycerol were added to the pork chops. The information on the form in which salt and glycerol were added to the meat (injection, pouring, dry?) should be completed.

In lines 101-102 it is stated that after sonication the samples were marinated .... (no information what was the composition of the marinade?).

In line 129, the information about the dimensions of the T36 probe should be completed.

In line 131, I suggest replacing "pressure" with "load".

Reviewer 3 Report

This is very interesting and original approach to the problem of enhancement of quality of products with reduced salt content. It would be interesting to see how would this technique authors propose work in the production of some dry/cured products like ham and even some dry fermented sausages. It would also be good that the authors put some references about this aspect.

Author Response

Thank you very much for giving us this opportunity to revise our manuscript. Those comments are all valuable and very helpful for revising and improving our manuscript. We also appreciate the time and effort the three anonymous reviewers have dedicated to providing insightful feedback on ways to strengthen our manuscript.

Yours regards,

Qiujin Zhu